# Chinee Apple (*Ziziphus mauritiana*): A Comprehensive Review of Its Weediness, Ecological Impacts and Management Approaches

**DOI:** 10.3390/plants12183213

**Published:** 2023-09-08

**Authors:** Ciara J. O’Brien, Shane Campbell, Anthony Young, Wayne Vogler, Victor J. Galea

**Affiliations:** 1School of Agriculture and Food Sustainability (AGFS), Gatton Campus, The University of Queensland, Gatton, QLD 4343, Australia; shane.campbell@uq.edu.au (S.C.); anthony.young@uq.edu.au (A.Y.); v.galea@uq.edu.au (V.J.G.); 2Department of Agriculture & Fisheries, Tropical Weeds Research Centre, P.O. Box 976, Charters Towers, QLD 4820, Australia; wayne.vogler@daf.qld.gov.au

**Keywords:** Chinee apple, Indian jujube, *Ziziphus mauritiana*, woody weeds, invasive alien species

## Abstract

*Ziziphus mauritiana* Lam. (Rhamnaceae) (Chinee Apple, Indian Jujube, or Ber) is a significant woody weed in the drier tropics of northern Queensland, Western Australia, and the Northern Territory. Throughout these regions, its densely formed thickets influence the structure, function, and composition of rangeland ecosystems by outcompeting native pasture species. Despite this, the recent literature is heavily focused on the horticultural value of domesticated *Ziziphus* species in South Asia (China, India, and Pakistan), particularly its potential for poverty alleviation in arid or semi-arid areas. In fact, there has been comparatively little research undertaken on its invasiveness or associated ecological factors in pastoral contexts. Currently, the management of *Z. mauritiana* is limited to the application of synthetic herbicides or mechanical clearing operations. There is also considerable interest in the exploitation of host-specific, natural enemies (biological control agents, herbivorous insects, fungi, bacteria, or viruses) for limiting the vigour, competitiveness, or reproductive capacity of *Z. mauritiana* in northern Australia. The development of a “bioherbicide” in lieu of synthetic counterparts may foster a more resilient coexistence between agricultural systems and the natural environment owing to its reduced environmental persistence and increased target specificity. This review summarises the current literature on the weediness, ecological impacts, and current management of this problematic weed, thereby identifying (i) opportunities for further research and (ii) recommendations for improved management within its invasive range.

## 1. Introduction

*Ziziphus mauritiana* Lam. (Rhamnaceae) (Chinee apple, Indian jujube, or ber) is a deciduous thorny tree or shrub native to South Asia (China, India, and Pakistan) and eastern Africa [1,2,3,4,5], where it has a remarkable history of ancient cultivation (~3200 years). More recently, it has served as a crucial source of food security for poor and resource-scarce populations in these arid areas [6]. However, it has also established invasive populations in northern Australia [1,5], southern Africa (i.e., Zimbabwe and Zambia) [7,8], Fiji [7], and some Pacific and Indian Ocean islands [7,9]. In Australia, it was introduced to the early mining settlements of northern Queensland (e.g., Charters Towers, Ravenswood, Mingela, and Hughenden) in the late nineteenth century for its ornamental and horticultural value [1,2,3,5,7]. Its current distribution is densest in the northern parts of Queensland (Townsville–Charters Towers), Western Australia (Broome, Derby, and Kimberley), and the Northern Territory (Darwin, Daly River, and Roper River Catchment) [1,3,5]. Throughout these regions, its densely formed thickets influence the structure, function, and composition of rangeland ecosystems by outcompeting native pasture species [3,5,10]. This negatively affects the quality of services and pastoral operations (e.g., agronomic productivity, livestock carrying capacity, water accessibility, and mustering) obtainable from this diverse, natural resource [3,5,7,11,12]. Hence, it has been identified as a priority threat to the pastoral industry by both graziers and landholders [1,5,7]. The biodiversity of tropical woodlands is also threatened, whereby the herbaceous understorey of native vegetation is replaced by a dense layer of impenetrable thickets [3,13]. The transformation of these habitats is associated with the decline in or localised extirpation of endemic wildlife [14,15], such as the threatened black-throated finch (southern subspecies: *Poephila cincta cincta*) in north-eastern Queensland [15].

The management of *Z. mauritiana* is limited to the application of synthetic herbicides or mechanical clearing operations [3,5,7]. Manual removal of lower density (<50 plants/ha) or isolated infestations can be achieved by stick-raking, blade ploughing, or bulldozing of individual trees [5,16]. However, there are obvious limitations in terms of cost efficiency, labour supply, and its effectiveness on medium-scale (50 to 150 plants/ha) or large-scale populations (>150 plants/ha) [5,17,18]. The basal or cut-stump application of synthetic auxin herbicides (Group 4: triclopyr, fluroxypyr, or picloram) is the primary means for the effective suppression of *Z. mauritiana* in northern Australia [5,16]. Whist their efficacy is undisputed, there are concerns regarding the suitability of synthetic herbicides in ecologically sensitive or low-value habitats [19]. Their excessive or improper application is associated with adverse injury to neighbouring plant communities via herbicidal spray drift, runoff, or leaching. This recent appreciation for environmental stewardship has promoted significant developments in the field of woody weed management by reducing dosage levels or improving application methods [5,19]. There is also considerable interest in the exploration of host-specific, natural enemies for limiting the vigour, competitiveness, and reproductive capacity of *Z. mauritiana* in northern Australia [3,7].

The recent literature is heavily focused on the horticultural value of domesticated *Ziziphus* species (*Z. mauritiana* and *Ziziphus jujuba* Mill.) in South Asia [1,13] and its potential for poverty alleviation in arid or semi-arid climates [6,20,21]. There has been comparatively little research undertaken on its invasiveness in northern Australia [13]. In fact, a comprehensive review of *Z. mauritiana* has not been published since that by Grice (2002) [22], with the exception of a recent examination focused solely on prospective biological control agents [6]. The primary objective of this review is to summarise the current literature on the species’ weediness, ecological impacts, and management to identify knowledge gaps that could be the catalyst for further research on this problematic weed.

## 2. Historical and Global Significance of *Ziziphus mauritiana* Fruit Trees

The “ber” fruit (*Z. mauritiana*) has an ancient history of cultivation throughout the plains of northern India [23,24,25]. Perhaps the earliest reference to this fruit is in the *Yajurveda* (c. 1200–800 BCE), a primary religious *Veda* text [23,24,25]. The primary region of historical cultivation, the Deccan Plateau, also predated the Gangetic civilisation (c. 1500–500 BC) [25]. Much later in the late eighteenth century, the fruit was presented to King Raghoji Bhonsle II of the Kingdom of Nagpur by a local Muslim farmer, thereby cementing its popularity in present day India [26]. Ultimately, this historical trajectory underscores its cultural and horticultural significance in most of South Asia [26]. In India, it has been successfully cultivated in the arid and semi-arid zones, particularly Andhra Pradesh, Bihar, Gujarat, Haryana, Madhya Pradesh, Maharashtra, Punjab, Rajasthan and Uttar Pradesh [26,27,28,29,30,31]. It is considered the “king of arid-zone fruits” or the “poor man’s apple” because of its lowered cost of production, high-yielding nature, drought and salinity tolerance, nutritional value, and scope for value addition (e.g., beverages, jams, cake, bread, and porridge) [20,21,29]. In fact, the nutritive richness (i.e., ascorbic acid, vitamins, minerals, and polyphenols) of “ber” fruit has been well documented in the current literature [6,21,23,24,26,32] by Muhammad et al. (2022) [6], Prakash et al. (2020) [32], and Sharif et al. (2022) [21]. Several studies have also demonstrated the pharmacological potential of all parts of *Z. mauritiana* as an antioxidant, antimicrobial, antidiarrheal, antidepressant, immunomodulator, and hepatoprotective [25,32,33,34,35,36] (Table 1). These therapeutic properties are attributable to a diverse assortment of derivative metabolites, such as flavonoids, alkaloids, terpenoids, glycosides, and saponins [21,33,34,36].

*Ziziphus mauritiana* is also cultivated on a smaller scale in other south Asian (Bangladesh, Bhutan, Nepal, Pakistan, and Sri Lanka), central Asian (Afghanistan, Iran, Iraq, and United Arab Emirates), and Arabian Gulf countries [24,25,26]. In these countries, the exacerbation of climate change has resulted in the expansion of arid land by altering precipitation patterns and intensifying drought conditions [6]. A recent study by Spinoni et al. (2021) predicted a significant climatic shift (~1.1% to 3.8%) in arid or semi-arid zones under four realistic “global warming levels” [38]. Therefore, the recent literature is heavily focused on domesticated *Ziziphus* varieties as a potential economic source in salt-affected soils or water-stressed conditions [6,29].

Although it has no commercial value in Australia [7], another *Ziziphus* species (Chinese jujube: *Z. jujuba*) has market potential in the southern states (New South Wales, South Australia, and Victoria) and southern parts of Western Australia [7,39,40,41]. Its commercial cultivation is mostly distributed throughout the Northern Rangelands, Perth Hills, and south-western region of Western Australia [39,40,41], where it is ecologically distinct from the invasive populations of *Z. mauritiana* in the northern states [7]. The production of “jujube” is a diversification opportunity for these farmers to build business resilience in the event of unpredictable salinity or seasonal drought [42]. Additionally, the proximity of these growing regions to South Asia creates a favourable scenario for the development of an export market to meet “off-season” demand in those countries [39,40,42]. For example, the largest producer of kiwifruit (*Actinidia deliciosa*) in Australia and New Zealand has recently invested in the production of commercial “jujube” varieties with the intention of launching an exportation industry [43].

## 3. Botany, Biology, and Ecology of *Ziziphus mauritiana*

*Ziziphus mauritiana* is a single- or multi-stemmed tree (height of <15 metres) with an intricately branched, spreading canopy (Figure 1A) [7,27]. The younger stems are covered in densely interwoven, woolly hairs and contain a single, curved thorn at each joint [22]. The upper side of its alternate, sub-elliptically shaped leaves (20 to 80 mm length) are dark-green, glossy, and glabrous (Figure 1B) [7,22], whilst the lighter-coloured underside is covered in white to rusty fine hairs (Figure 1D) [22]. Its small, inconspicuous flowers (5–8 mm width) are greenish-yellow with a hypanthium floral structure [7,27]. They are clustered in cyme inflorescences of twelve to fourteen flowers that are connected by short pedicles (<4 mm) [27]. The floral biology and development of *Z. mauritiana* has been described in further detail by Tel-Zur et al. 2009 [27].

The sub-globular, drupaceous fruit (20–50 mm diameter) have a leathery exocarp that varies in colouration between yellow-green and reddish-brown as an indication of ripeness (Figure 1C) [3,22,26]. They also have a lignified, irregularly furrowed endocarp surrounded by a white fleshy mesocarp that is palatable [3,26]. The intact endocarp encases one, sometimes two or three, rounded darker brown seeds (~6 mm) [1,3,7,13]. Despite this mechanism of physical dormancy, the seed viability and persistence are relatively short-lived (<2 years) in the soil [3,7]. In fact, Grice (1996) discerned that less than 10% of seeds at the soil surface or a shallow burial depth (~2 cm) were germinable after only twelve months [1]. However, the fecundity of *Z. mauritiana* has also been well documented in the literature [3,13]. In a single reproductive event, the production of more than 5000 seeds is typical of larger shrubs [1,3,7].

The consumption of seed by frugivorous birds is a potential avenue of localised dispersal [1,3]. A study by Grice (1996) observed the transportation of seed by pied currawongs (*Strepera graculina*), red-tailed black cockatoos (*Calyptorhynchus banksii*), and channel-billed cuckoos (*Scythrops novaehollandiae*) in the tropical woodlands of northern Australia [1]. They commonly feed on mature fruits in the canopy or harvest seed from naturally fallen fruits on the soil surface [1]. The partial consumption of fruit flesh has also been recorded in pale-headed rosellas (*Platycerus adscius*) and red-winged parrots (*Aprosmictus erythropterus*) [1,22]. However, the importance of avian seed transportation is largely unknown because of the lack of research on (1) their territorial behaviours or boundaries, (2) the time for seed material passage, and (3) the potential disgorgement of larger, woody endocarps [13].

Many studies have found that mammals likely have a significant role in the translocation of *Z. mauritiana* seeds. In Queensland, a large number of intact endocarps (<240 and µ = 17) [27] with viable seeds were collected from the faeces of domestic cattle (*Bos indicus*), feral pigs (*Sus scrofa*), and native agile wallabies (*Notamacropus agilis*) [1,13,22]. This is associated with the wider movement of seed material to nearby sites or over several kilometres [1,22]. Other mammals and birds are also involved in seed transportation, such as horses (*Equus ferus*), donkeys (*Equus asinus*), camels (*Camelus dromedaries*), goats (*Capra aegagrus hircus*), sheep (*Ovis aries*), emus (*Dromaius novaehollandiae*), and bustards (*Ardeotis australis)* [3]. Additionally, many seeds remain beneath or very close to the canopy of the parent plant [1,13]. This is evidenced by the frequent establishment of juvenile seedlings around reproductively mature plants at infestation sites throughout northern Australia [1].

*Ziziphus mauritiana* is a very hardy tree tolerant to extreme temperature variability (−5 °C to 49 °C) and dry conditions [7,23,27,29,30,31,44] and therefore is well suited to the seasonally variable rainfall patterns of northern Australia (average annual rainfall of 500 mm to 1500 mm) [45]. It is also successfully cultivated in the arid and semi-arid zones of north-western India (e.g., Gujarat, Haryana, and Rajasthan) [29,31], where the annual rainfall can be as low as 200 mm [30]. Clifford et al. (1998) investigated the physiological basis of drought tolerance in *Z. mauritiana* under glasshouse conditions [30]. This study found that a combination of solute accumulation and increased cell rigidity were the likely mechanisms for drought tolerance in this particular species [30]. However, a subsequent study by Arndt et al. (2000) indicated that it also accessed moisture deeper within the soil profile via the taproot system [46]. There are no specific soil requirements for *Z. mauritiana* throughout its native or naturalised distribution [7,22,44]. In fact, its successful establishment has been cited in deep coarse-textured sands, shallow-surfaced solodic soils, cracking clays, deep alluvials, and skeletal soils [22,23,44].

## 4. Native and Naturalised Distribution of *Ziziphus mauritiana*

*Ziziphus mauritiana* is native to South Asia (China, India, and Pakistan) and eastern Africa [1,2,3,4,5] (Figure 2). However, it has established invasive populations in northern Australia [1,5,44], southern Africa (i.e., Zimbabwe and Zambia) [7,8], Fiji [7], and on some Pacific and Indian Ocean islands (e.g., Seychelles) [7,9,44] (Figure 2). Its naturalisation has also been recorded in the arid or semi-arid zones of South America [7,37,44], Central America [7,37], United States of America [7,37,44,47], Western Africa, the Middle East [22,37], the Caribbean [44], Cape Verde Islands, and La Reunion [7] (Figure 2).

In Australia, the densest populations of *Z. mauritiana* are associated with former mining settlements (e.g., Charters Towers, Hughenden, Mingela, and Ravenswood) in northern Queensland [1,2,3,22], where it was initially introduced in the late nineteenth century (1863) for its horticultural value [1,2,3,5,7,22,44] (Figure 3). A study by Grice et al. (2000) examined the regional and landscape patterns of this invasive shrub in the Charters Towers region (area of 68,388 km^2^) of northern Queensland [2]. It was recorded in 32% of sites within a 20 km radius of Charters Towers [2]. A similar pattern of occurrence has been documented in the Northern Territory (e.g., Darwin, Daly River, and Rope River Catchment) and Western Australia (e.g., Broome, Derby, Kimberley, and Kununurra) [1,22], as shown in Figure 3.

A CLIMEX model was developed by the Queensland Government for estimation of the potential distribution or relative climatic suitability of *Z. mauritiana* in Queensland (Figure 4) [52]. The entire eastern coast of Queensland (from Brisbane to Cape York Peninsula) was deemed highly suitable for invasion by this species, with a high probability of occurrence predicted in most of the dry and wet tropics (Figure 4). A similar distribution was also documented in parts of south-central Queensland, as well as west of Rockhampton (Figure 4). A lowered suitability was predicted further inland towards central-west and north-west Queensland (Figure 4). These species distribution models (SDMs) are valuable in the identification of prospective locations for the exploration or targeted establishment of biological control agents [53,54,55].

## 5. Agricultural, Environmental, and Economic Impacts of *Ziziphus mauritiana*

In Australia, this species is a problematic weed in both grassland and rangeland environments [22]. Its densely formed thickets influence the structure, function, or composition of rangeland ecosystems by outcompeting native pasture species [3,5,7,10,44]. This negatively affects the quality of services and pastoral operations obtainable from this diverse, natural resource, such as the agronomic productivity, livestock carrying capacity, water accessibility, and mustering activities of the land [3,5,7,12]. Hence, it has been identified as a priority threat to the pastoral industry by graziers and other landholders [1,5,7]. There has been no quantitative assessment of the economic costs associated with *Z. mauritiana* in pastoral contexts [22]. The realisation of these costs will assist in (1) the prioritisation of research efforts, (2) resource allocation, (3) policy formation, (4) stakeholder collaboration, and (5) the development or rationalisation of management decisions [56,57].

From a conservation perspective, the biodiversity of tropical woodlands is also threatened by *Z. mauritiana* [3,13]. These habitats of scattered eucalypts are vulnerable to invasion by several invasive shrubs, particularly Chinee apple (*Z. mauritiana*), parkinsonia (*Parkinsonia aculeata*), prickly acacia (*Vachellia nilotica*), mesquite (*Prosopis* spp.), and rubber vine (*Cryptostegia grandiflora*) [58], whereby the herbaceous understorey is replaced by a dense layer of impenetrable thickets [15,58]. The transformed architecture of these woodlands is associated with the localised extirpation of or reduction in native wildlife [14,15]. For example, a recent study by Laguna et al. (2019) found that the proliferation of two invasive shrubs (*Z. mauritiana* and *Lantana camara*) in grassy woodlands coincided with a declined population in the endangered black-throated finch (southern subspecies: *Poephila cincta cincta*) [15].

In Queensland, *Z. mauritiana* is a Category 3 restricted species under the *Biosecurity Act 2014* [59], meaning that the introduction, release, or commercial use of any plant matter is prohibited without an authorised permit [7,59]. Similarly, this invasive shrub is declared a Class A (i.e., must be eradicated) and Class C (i.e., cannot be introduced) weed in the Northern Territory as part of the *Weeds Management Act 2001* [22,60]. The eventual eradication of *Z. mauritiana* is deemed feasible because of its limited or isolated establishment in Darwin, Daly River, and the Rope River Catchment. The only other state with a current declaration status is Western Australia [22,61], whereby some form of management is required (C3) under the *Biosecurity and Agriculture Management Act 2007* [61].

Currently, *Z. mauritiana* is more problematic in agricultural production contexts, particularly in extensive livestock systems. This means that there are very few examples of its control in other situations (e.g., tropical woodlands or forestry environments). However, the management of many invasive woody weeds amongst native vegetation must be undertaken in compliance with vegetation management regulations, such as the *Vegetation Management Act 1999* in Queensland [62], whereby the conservation and biodiversity of remnant vegetation is prioritised [63]. This can be achieved via the establishment of Area Management Plans (AMPs) by natural resource management organisations [62]. For example, the treatment of *Z. mauritiana* in environmentally sensitive habitats is approved and outlined under the “Dry Tropics Weed AMP” of northern Queensland [64].

The “dual nature” of most weeds is either rarely acknowledged or poorly understood in the current ecological literature [14,65]. However, some recent studies have demonstrated that the interaction between native biodiversity and exotic invasion is not always inevitably negative [14,65,66], a phenomenon referred to as “the invasion paradox” [62,63]. For example, Ward-Fear et al. (2017) observed that *Z. mauritiana* offered critical refuge to native rodents (pale field rats: *Rattus tunneyi*) from feral horses (*E. ferus*) in the remote floodplains of north-western Australia [14]. The spiny thorns excluded feral horses from the shaded area beneath the canopy, thereby providing the only sites for rat burrows within the landscape that were not subject to trampling and soil compaction [14]. Although this ecosystem is under simultaneous threat, this invasive shrub essentially buffers the impacts posed by the feral horses and subsequently enhances the survival of the threatened rodents [14]. Therefore, the eradication of *Z. mauritiana* in this degraded ecosystem may in fact cause population bottlenecks, local extinction, or trophic cascades [14,65].

## 6. Management of Invasive *Ziziphus mauritiana* in Australia

### 6.1. Manual and Mechanical Control Methods

The manual removal of lower density (<50 plants/ha) or isolated infestations can be achieved through stick-raking, blade ploughing, or bulldozing of individual trees [5,16]. The most suitable period for these activities is prior to fruit development or lowered root reserves [16]. However, there are obvious limitations in terms of cost efficiency, labour supply, and effectiveness for larger-scale populations [5,17,18]. For example, a blade plough (depth of 150–200 mm below ground level) is only effective on plants with a substantial root system, yet not too oversized for the capacity of the machinery [67]. The vigorous resprouting of roots or lignotubers is also highly likely following some types of mechanical disturbance (e.g., bulldozing), whereby the plant is severed at the base [16,22]. Therefore, the treatment of subsequent regrowth or exposed stems with synthetic herbicides is often needed [16,22].

In Australia, the estimated annual cost of woody weeds to the grazing industry is in the vicinity of AUD 12.3 billion [68]. Although a “trade-off” exists, the management of *Z. mauritiana* is largely very costly relative to the annual returns of rangeland environments (Table 2) [69,70]. A study by Zull et al. (2008) constructed an analytical framework for the optimal frequency of management by synthesising the complex relationships between population dynamics, direct weed costs, and the cost–benefit of different control methods [70]. This model suggested that mechanical means of management were largely uneconomic for *Z. mauritiana*, particularly in upland zones [70].

The encroachment of many invasive woody weeds can be controlled with “prescribed fire” either directly or as part of an integrated approach [71,72]. Some successful examples of this include rubber vine (*C. grandiflora*), bellyache bush (*Jatropha gossypiifolia*), mesquite (*Prosopis* spp.), parkinsonia (*P. aculeata*), lantana (*L. camara*), and prickly acacia (*V. nilotica*) [72]. A study by Grice (1997) investigated a controlled fire event on the survival and vegetative growth of *Z. mauritiana* in northern Queensland [58]. Although considerable mortality (>90%) has been previously observed in surface-located seeds [73], the survival rate of established plants was similar in burnt versus unburnt plots [58]. In fact, most plants resprouted vigorously within three months of fire treatment, even those of smaller height classes (<100 cm) [58]. Anecdotal reports suggest that they recover with increased vigour, similarly to other fire-tolerant woody weeds (e.g., calotrope: *Calotropis procera* and Gorse: *Ulex europaeus*). However, there is an opportunity for further research in the application of chemical defoliants following a single or repeated fire event [12,58,74].

### 6.2. Chemical Control Methods

Although costly, the basal or cut-stump application of synthetic auxin herbicides (Group 4: triclopyr, fluroxypyr, or picloram carried in diesel) is the most effective means for the aggressive eradication of higher density populations (>150 plants/ha) [5,16,60,61]. There are several chemical products registered for the management of *Z. mauritiana* in northern Australia [16,22,75,76], as shown in Table 3. In particular, the basal bark application of triclopyr (600 g/L), fluroxypyr (333 g/L), or a combination herbicide (triclopyr 240 g/L + picloram 120 g/L) is suitable for seedlings and juvenile plants in their active growth stage (Table 4) [16,22]. Alternatively, the same herbicide mixtures are also recommended for the treatment of cut-stumps at any time of year (Table 3) [16,22]. However, these methods may be constrained by the spiny lower branches or multi-stemmed character of some shrubs [22]. Therefore, a high-volume spray mixture in water (350 mL herbicide mixture to 100 L water) can be applied to actively growing juvenile or established plants (Table 3) [16,22].

The efficacy of synthetic herbicides is often influenced by the growth or reproductive cycles of the target species at the time of application [16,77]. For example, the sensitivity of some weeds is increased in the active reproductive stages (e.g., flowering or fruiting) [16] (Table 4). In this instance, the production of viable fruit is disrupted, thereby limiting the further spread and establishment of the weed (Table 4) [16]. Similarly, the treatment of seedlings or juvenile plants before the first seed set (e.g., the first two to five years) can inhibit critical metabolic and developmental processes within the plant (Table 4) [16]. The ideal time of varying herbicide application methods for *Z. mauritiana* is summarised in Table 4.

A proprietary stem implantation system has been developed by BioHerbicides Australia (BHA Pty Ltd.) for the encapsulated delivery of an endophytic fungal bioherbicide (*Lasiodiplodia pseudotheobromae*, *Macrophomina phaseolina*, and *Neoscytalidium novaehollandiae*) in parkinsonia (*P. aculeata*) [78,79,80]. This novel technology has since been expanded to the application of other endophytic organisms, as well as synthetic herbicide compounds available in dry formulations [5,79,81,82]. Several synthetic herbicide formulations have been trialed for the management of various invasive woody weeds: calotrope (*C. procera*) [81], camphor laurel (*Cinnamomum camphora*) [81], Chinese elm (*Celtis sinensis*) [18], leucaena (*Leucaena leucocephala*) [81], prickly acacia (*V. nilotica*) [81], and mimosa bush (*Vachellia farnesiana*) [82]. Recently, preliminary research was undertaken by O’Brien et al. (2022) on the compatibility of this technology with *Z. mauritiana* in rangeland environments [5]. The initial results are very promising, whereby three of the encapsulated synthetic herbicides (Di-Bak AM^®^: aminopyralid + metsulfuron-methyl, Di-Bak^®^ M: metsulfuron-methyl, and Di-Bak P^®^: picloram) achieved a similar response to the “drill-and-fill” application of Tordon^®^ RegrowthMaster (triclopyr, picloram, and aminopyralid) [5]. However, unlike its industry counterparts, a minimum recommended lethal dose is delivered directly into the vascular system, where the respective active ingredient is fully captured internally [5,79,81]. This targeted, readily calibrated herbicide application is associated with (1) a lowered active ingredient concentration (~20% to 30%), (2) a reduced likelihood of environmental exposure to plant protection compounds, and (3) improved safety for the human operator [79]. This technology could be a possible replacement for conventional methods of foliar or stem spraying, stem injection, or canopy application in environmentally sensitive habitats (e.g., tropical woodlands, native forestry, and riparian zones) [79].

### 6.3. Biological Control Methods

There is considerable interest in the exploitation of host-specific, natural enemies for limiting the vigour, competitiveness, and reproductive capacity of *Z. mauritiana* in northern Australia [3,7]. The development of a “bioherbicide” in lieu of their synthetic counterparts may foster a more resilient coexistence between agricultural systems and the natural environment owing to their reduced environmental persistence and increased target specificity [71,83]. It is noted that they are not a total replacement or “panacea” to conventional weed management [83]. Rather, they could be used concurrently with other control methods (e.g., synthetic herbicides or mechanical options) as part of a wider integrated approach with multiple modes of action [83].

Currently, *Z. mauritiana* has not been the focus of any biological control programs [3,7,22], although there is a plethora of published literature available on the pests and diseases of *Ziziphus* species [6]. In fact, Dhileepan (2017) explored the feasibility of biological control by cataloguing the phytophagous insects, mites, and pathogens of two wild *Ziziphus* species (*Z. mauritiana* and *Z. jujuba*) within their native range [7]. These opportunistic field surveys and literature searches identified a suite of prospective agents with differing feeding guilds, including the leaves, shoots, stem, fruit, and seed [7]. Of these, there were seven phytophagous arthropods (i.e., four leaf-feeding, two shoot-feeding, and one seed-feeding arthropods) with potential host specificity for *Z. mauritiana* (Table 5) [7]. In particular, a seed-feeding weevil (*Aubeus himalayanus*) and two shoot-galling mites (*Aceria cernuus* and *Larvacarus transitans*) were considered the most suitable biological control candidates for northern Australia [7].

Among all the known fungal pathogens of *Ziziphus* species, leaf rust (*Phakopsora zizyphi-vulgaris* Diet.) and powdery mildew (*Pseudoidium ziziphi*) have the most restricted host specificity [7]. Autoecious rust (*P. zizyphi-vulgaris*) is a significant disease of commercial jujube (*Z. jujuba*) in China [84], associated with the formation of irregular reddish-brown pustules on the entire leaf surface and eventual defoliation [7,84]. In India, it has also been recorded in wild varieties of *Z. mauritiana*, wild jujube (*Ziziphus nummularia*), and jackal jujube (*Ziziphus oenoplia*) [7]. The latter is a native tree of environmental significance in northern Australia, and therefore *P. zizyphi-vulgaris* is an unsuitable candidate for biological control [7]. There are only two recorded hosts (*Z. mauritiana* and *Z. nummularia*) of powdery mildew (*P. ziziphi*) in India, Pakistan, and Bangladesh [7,84]. The younger leaves are often covered in a white powdery mass followed by shrinking and eventual defoliation [84,85]. The fruits are also significantly affected (yield loss of ~50% to 60%), whereby they become corky, cracked, or underdeveloped [84,85]. Of the two diseases, powdery mildew should be the focus of future surveys based on its host specificity and severity [7].

There is no current information on the susceptibility of various plant parts and growth stages (i.e., seedlings, juvenile, or adult plants) of *Z. mauritiana* to herbivory or diseases [7]. Furthermore, the literature available on its natural enemies is focused mostly on cultivated trees, and further research should be undertaken on the surveying of wild *Z. mauritiana* [7]. There are also two native *Ziziphus* species (*Ziziphus quadrilocularis* and *Z. oenoplia*) of environmental significance in northern Australia [7]. Therefore, the host specificity of any prospective agent(s) should be reviewed extensively to avoid deleterious impacts on valuable native flora [7].

## 7. Conclusions

The recent literature is heavily focused on domesticated *Ziziphus* species (*Z. mauritiana* and *Z. jujuba*) as a potential economic source in salt-affected or water-stressed environments [6,20,21,86]. Despite its overt status as a problematic weed, there has been comparatively little research undertaken on its invasiveness or related ecological factors in other landscapes [13]. The management of *Z. mauritiana* is currently restricted to the application of synthetic herbicides or mechanical clearing operations [3,5,7]. However, they have limited effectiveness individually, and a more integrated weed management (IWM) approach is recommended [83,87]. There is also considerable interest in the expansion of control options through the exploration of host-specific, natural enemies for limiting the vigour, competitiveness, and reproductive capacity of many woody weeds, including *Z. mauritiana* [3,7]. To move forward with these biological control efforts, it is essential for future research to delve into the introduction history of *Z. mauritiana* in Australia [7]. The genetic diversity and relatedness of differing populations or biotypes (native, naturalised, and invasive ranges) should also be determined [7]. Furthermore, a species distribution model (SDMs: CLIMEX or MaxEnt) of the other northern states in Australia (i.e., Western Australia and the Northern Territory) is recommended to assess the relative climatic suitability of specific locations for the exploration or targeted establishment of biological control agents [7,53,54,55]. Although not yet published, the pathogenicity of several fungal endophytes recovered from healthy and diseased populations of *Z. mauritiana* in northern Queensland is under current assessment as prospective “bioherbicide” candidates.

## Figures and Tables

**Figure 1 plants-12-03213-f001:**
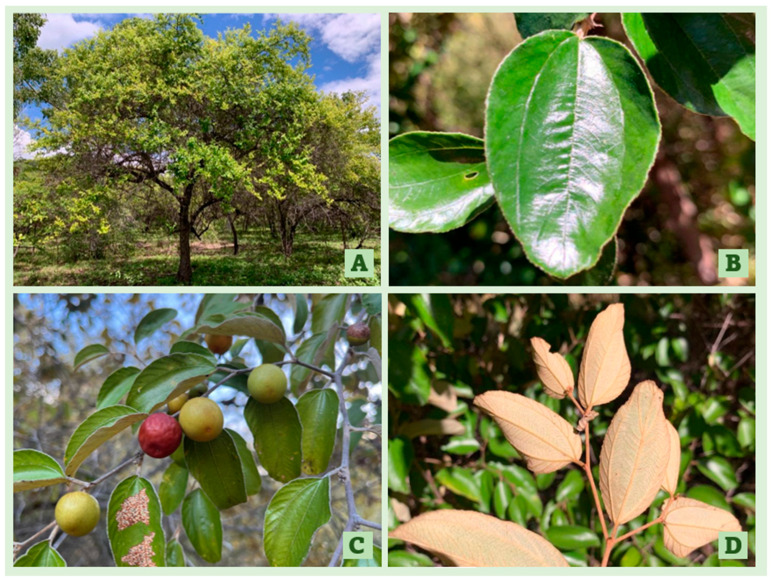
The invasive *Ziziphus mauritiana* in Northern Queensland, Australia: (**A**) branched, spreading canopy, (**B**) upper side of the leaf, (**C**) sub-globular, drupaceous fruit of varying degrees of ripeness, and (**D**) underside of the leaf.

**Figure 2 plants-12-03213-f002:**
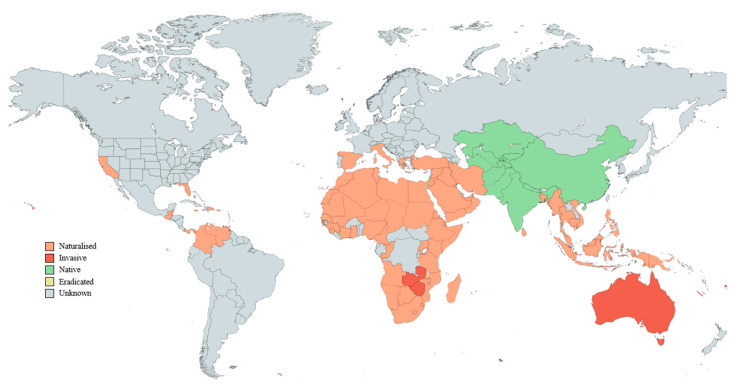
A presence-only global distribution map of *Ziziphus mauritiana* based on species’ records from the Centre for Agriculture and Bioscience International (CABI) [48], World Agroforestry Centre [49], Pacific Islands Ecosystems at Risk (PIER) [50], and Morton (1987) [37].

**Figure 3 plants-12-03213-f003:**
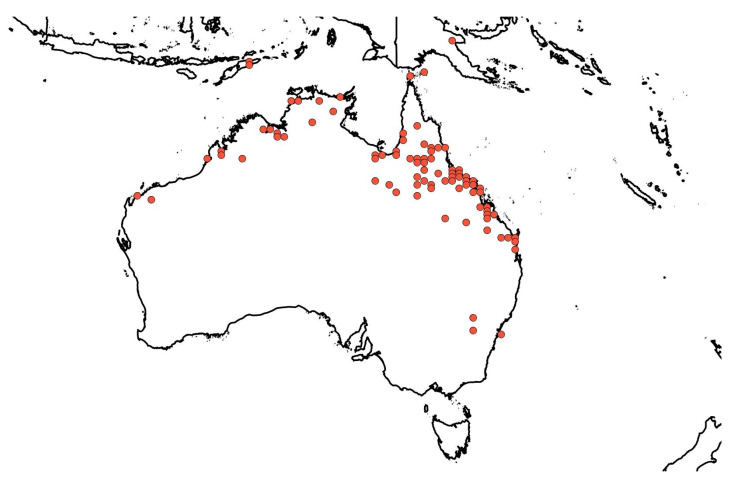
The distribution of *Ziziphus mauritiana* in northern Australia from the national and state herbarium occurrence records (~332 records from 1909 to 2019) of New South Wales, Queensland, South Australia, Victoria, Western Australia, and the Northern Territory [51]. Its presence in Timor-Leste has also been documented [51].

**Figure 4 plants-12-03213-f004:**
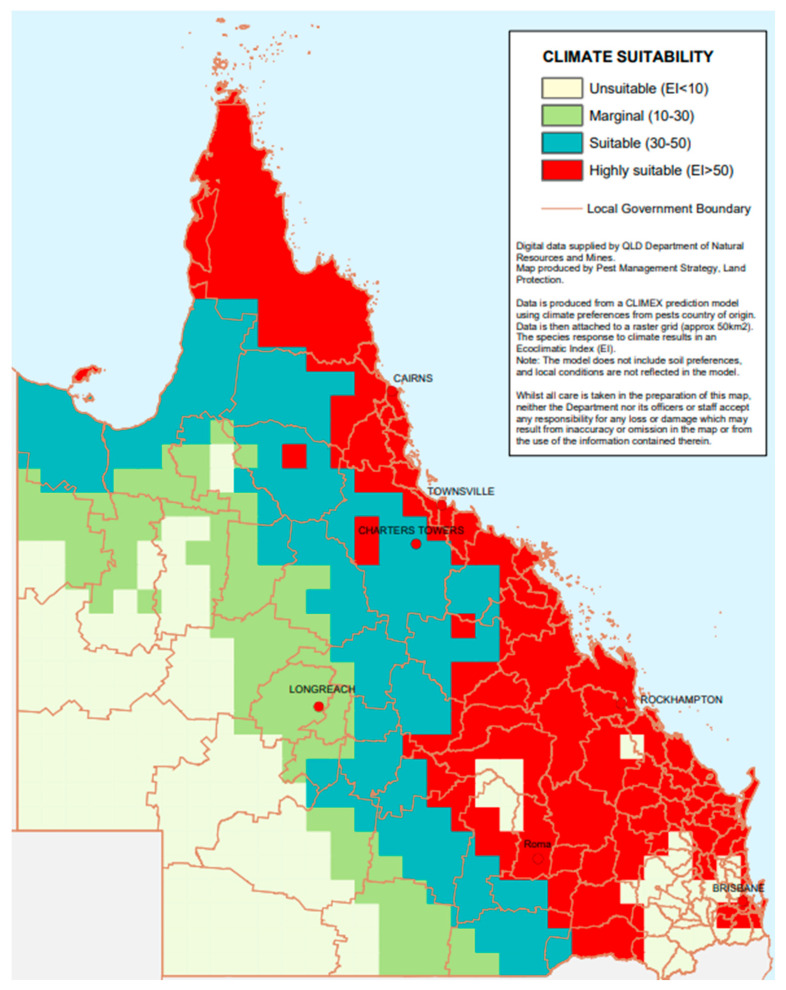
A CLIMEX prediction model of the potential distribution or relative climatic suitability of *Ziziphus mauritiana* in Queensland, Australia. The data are attached to a raster grid (approx. 50 km^2^) [52].

**Table 1 plants-12-03213-t001:** The pharmacological potential of all parts (fruit, seed, leaves, bark, and roots) of *Ziziphus mauritiana* for the treatment of various ailments or diseases. The data were sourced from Morton 1987 [37], Meghwal et al. 2007 [23], Naaz et al. 2020 [33], and Butt et al. 2021 [36].

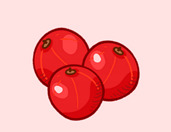	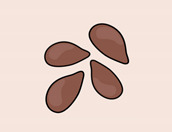	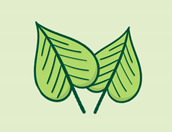	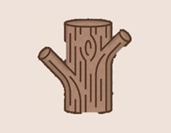	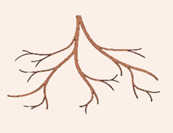
FRUIT	SEED	LEAVES	BARK	ROOTS
Blood Disorders Constipation Dyspepsia Fevers Flatulence Hyperdipsia Indigestion Laxative Leprosy Ulcers Vomiting Wounds	Abdominal Pains Asthma Cough Diarrhea Encephalopathy Insomnia Nausea Ophthalmopathy Rheumatism Sedative	Asthma Blood Diseases Boils Conjunctivitis Diarrhoea Dysentery Fever Gum Bleeding Liver Issues Smallpox Stomatitis Syphilitic Ulcers Tuberculosis Typhoid Fever Wounds	Boils Diarrhoea Dysentery Gingivitis Ulcers	Analgesic Anti-Allergic Anti-Inflammatory Cephalalgia Dysentery Fever Gout Rheumatism Ulcers Wounds

**Table 2 plants-12-03213-t002:** The variable, fixed, and total cost of four different control methods (i.e., no control, prescribed burning, synthetic chemicals, and mechanical) for *Ziziphus mauritiana*. The variable costs are density-dependent, whereby additional materials and labour are required for very dense infestations [69].

	Fixed Costs (ha^−1^)	Variable Costs (ha^−1^)	Total Cost (ha^−1^)
No Control	AUD$0	AUD$0	AUD$0
Burning	AUD$15	AUD$0	AUD$15
Chemical	AUD$37.50	AUD$112.50	AUD$150
Mechanical *	AUD$50	AUD$50	AUD$100

* Blade-Ploughing.

**Table 3 plants-12-03213-t003:** The synthetic herbicides (trade names, active ingredient(s), and application rate) registered for the management of *Ziziphus mauritiana* in Queensland, Western Australia, and the Northern Territory [15,72,73].

	TRADE NAME	ACTIVE INGREDIENT(S)	APPLICATION RATE	
BASAL BARK & CUT-STUMP APPLICATION	Access^®^	Triclopyr (240 g/L) + Picloram (120 g/L)	1 L/60 L Diesel	GROUP 4 (*Disruptors of* *Plant Cell Growth*)
	Invader^®^ 600	Triclopyr (600 g/L)	1 L/60 L Diesel	
	Garlon^®^ 600			
	Redeem^®^ 600			
	Acclaim^®^	Fluroxypyr (200 g/L)	3 L/100 L Diesel	
	Flagship^®^ 200			
	Starane^®^ Advanced	Fluroxypyr (333 g/L)	1.8 L/100 L Diesel	
	Comet^®^ 400	Fluroxypyr (400 g/L)	1.5 L/100 L Diesel	
	Decoy^®^ 400			
HIGH VOLUME SPRAY	Fightback^®^	Triclopyr (300 g/L) + Picloram (100 g/L)	350 mL/100 L Water	
	Conqueror^®^			
	Grazon® Extra	Triclopyr (300 g/L) + Picloram (100 g/L) + Aminopyralid (8 g/L)	350 mL/ 100 L Water	
SOIL APPLICATION	Tordon^TM^ Granules	Picloram (20 g/kg)	35 to 45 g/m^2^	

**Table 4 plants-12-03213-t004:** A summary of the primary *Ziziphus mauritiana* reproductive events (i.e., flowering, fruiting, and germination) and their correlation with differing synthetic and manual management options in Northern Australia [16].

	JAN	FEB	MAR	APR	MAY	JUNE	JULY	AUG	SEP	OCT	NOV	DEC
FLOWERING	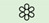	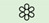	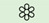	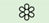	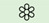							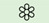
FRUITING	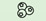	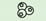	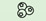	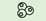	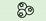	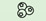	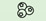	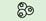				
GERMINATION											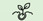	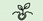
FOLIAR SPRAY	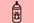	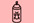	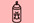	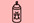	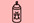						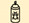	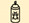
BASAL BARK	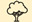	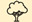	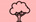	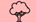	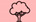	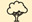	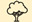	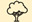	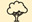	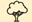	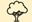	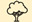
CUT-STUMP	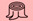	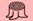	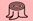	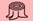	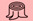	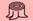	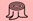	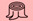	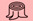	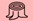	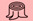	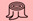
MANUAL	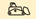	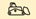	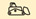	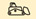	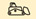	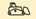	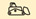	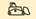	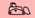	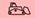	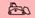	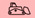

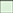
 Growth & Reproductive Events; 
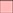
 Most Suitable Control Option; 
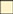
 Least Suitable Control Option.

**Table 5 plants-12-03213-t005:** The prospective agents with restricted host specificity identified by Dhileepan (2017) for the biological control of *Ziziphus mauritiana* in northern Australia [7].

Prospective Agent(s)	PHYTOPHAGUS INSECTS	
	Seed-Feeding Weevil	*Aubeus himalayanus*
	Leaf-Feeding Crambid Moth	*Synclera univocalis*
	Leaf-Feeding Gracillariid Moth	*Phyllonorycter iochrysis*
	Leaf-Galling Midge	*Phyllodiplosis jujubae*
	Stem-Galling Midge	*Silvestrina jujubae*
	PHYTOPHAGUS MITES	
	Shoot-Galling Mite	*Aceria cernuus*
	Shoot-Galling Mite	*Larvacarus transitans*
	FUNGAL PATHOGENS	
	Leaf Rust	*Phakopsora zizyphi-vulgaris*
	Powdery Mildew	*Pseudoidium ziziphi*

## Data Availability

Not applicable.

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
