# Peer review of "Chinee Apple (*Ziziphus mauritiana*): A Comprehensive Review of Its Weediness, Ecological Impacts and Management Approaches"

_plants, 2023, doi:10.3390/plants12183213_

Round 1

Reviewer 1 Report

In the review entitled " Chinee Apple (Ziziphus mauritiana): A Comprehensive Review of its Biology, Ecological Impacts and Management Approaches authors have discussed in detail the current literature on the biology, ecology, and management of Ziziphus mauritiana, through which they have unlocked opportunities for further research and various deliberations and suggestions for improved management within its invasive range.

In India Pakistan and some other southeastern regions, this invasive woody weed serves as a source of living for the poor and resource-scarce population.

How did you come up with a review on the less important crop in terms of it location in Australia, although it is popular in India? 

Review seems to be well-needed and timely and is comprehensive in nature.

Are there studies where the woody species is being dealt with manually and used as biofuel?

In the introduction section, u can add more literature on the use of the domesticated species of this weed as an alternate horticulture crop.

Throw some more light on the use of this plant as a potential economic source, particularly in salt-affected soils or heavy metal-contaminated soils.  Also, it grows under rainfed ecology, particularly in water-stressed conditions 

What are the steps being taken to manage the detrimental effects of biodiversity threat by the invasive woody weed species in Australia, particularly to other forest trees of economic importance... can be dealt with through detailed application reports from published sources .. In addition chemical methods.

bioherbicides 

role of plant biotechnology 

maybe added as a small section before concluding remarks  

Author Response

Thank you so much for your detailed feedback. Please see the attachment. 

Reviewer 2 Report

The present article entitled “Chinee Apple (Ziziphus mauritiana): A Comprehensive Review of its Biology, Ecological Impacts and Management Approaches” is based on a very nice theme. Author has covered a broad view of the Ziziphus mauritiana  in this article.

The article is well written and up to the mark. However I have some suggestion that may help to  improve the quality of the article.

Add a future perspective section that will focus on sustainable production or how to manage the loss of plants from the pathogen invasion. As now a days   next generation sequencing  have been used to explored the microbial  composition  and their effective role in plant disease management . So if possible, add these  points in the future perspective  section

The present article entitled “Chinee Apple (Ziziphus mauritiana): A Comprehensive Review of its Biology, Ecological Impacts and Management Approaches” is based on a very nice theme. Author has covered a broad view of the Ziziphus mauritiana  in this article.

The article is well written and up to the mark. However I have some suggestion that may help to  improve the quality of the article.

Add a future perspective section that will focus on sustainable production or how to manage the loss of plants from the pathogen invasion. As now a days   next generation sequencing  have been used to explored the microbial  composition  and their effective role in plant disease management . So if possible, add these  points in the future perspective  section

Author Response

Thank you so much for your feedback. Please see the attachment. 
